# Comparing Spatial and Temporal Variabilities between the Vaisala AQT530 Monitor and Reference Measurements

Roubina Papaconstantinou<sup>1</sup>, Spyros Bezzantakos<sup>1</sup>, Michael Pikridas<sup>1</sup>, Moreno Parolin<sup>1</sup>, Melina Stylianou<sup>2</sup>, Chrysanthos Savvides<sup>3</sup>, Jean Sciare<sup>1</sup>, and George Biskos<sup>1,4</sup>

<sup>1</sup>Climate and Atmosphere Research Centre, The Cyprus Institute, Nicosia 2121, Cyprus

<sup>2</sup>Medisell Co Ltd, Nicosia 2033, Cyprus

<sup>3</sup>Ministry of Labour and Social Insurance, Department of Labour Inspection, Nicosia 1463, Cyprus

<sup>4</sup>Faculty of Civil Engineering and Geosciences, Delft University of Technology, Delft, 2628-CN, the Netherlands

*Correspondence to:* George Biskos ([g.biskos@cyi.ac.cy](mailto:g.biskos@cyi.ac.cy))

**Abstract.** Low-cost gas and particle sensors can enhance the spatial coverage of Air Quality (AQ) monitoring networks in urban settings. While their accuracy is insufficient to replace reference instruments, they may still capture spatial differences among different stations, as well as temporal trends and month-to-month variabilities at a specific location. To assess this, we conducted a 19-month study using two Vaisala AQ Transmitters-Monitors (Model AQT530), collocated with reference-grade instruments, at two AQ stations in Nicosia: an urban traffic and an urban background station. These two stations are ideal for

the needs of this study considering that the reference measurements carried out there exhibit statistically significant spatial and temporal differences in pollutant concentrations when analysed over the entire period and on a monthly basis.

The AQT530 air quality monitor employs Low-Cost Sensors (LCSs) for gaseous pollutants (i.e., CO, NO<sub>2</sub>, NO and O<sub>3</sub>) and particulate matter (PM). Tests of the performance of the two AQT530 monitors during an initial period when those were collocated at the urban traffic station revealed high unit-to-unit agreements for the CO, NO and PM<sub>10</sub>, and good to moderate

for the NO<sub>2</sub>, O<sub>3</sub> and PM<sub>2.5</sub> measurements. The CO and PM<sub>10</sub> LCS measurements also effectively captured concentration differences between the two stations when averaged over the full study period or monthly, with some exceptions for specific months. These LCSs successfully detected spatial concentration differences (i.e., monthly, daily and hourly) as long as those were above a certain threshold. Overall, the CO and PM sensors successfully tracked month-to-month trends over the entire study period, similarly to reference instruments, whereas NO<sub>2</sub>, NO, and O<sub>3</sub> sensors struggled due to environmental sensitivities.

Despite this, all sensors identified statistically significant month-to-month variations at the same station, with PM<sub>2.5</sub> showing the strongest agreement with reference data.

## 1 Introduction

Air pollution is a major concern of our modern societies, due to its adverse effects upon human health and the environment (Juginovic et al., 2021; Kuntic et al., 2023). This is more so in urban agglomerates, where a range of human activities can yield 30 high concentrations of air pollutants at specific locations, typically referred to as air pollution hot spots, creating high spatial

and temporal variabilities within a city. Although capturing such variabilities is strongly desired, the high capital and maintenance costs of the necessary instruments still limit the density of urban air quality (AQ) monitoring stations. For example, in large European cities (e.g., London, Paris, Rome, Madrid, Berlin and Athens) the number of fixed AQ monitoring stations is of the order of 1 station every ca. 50 km<sup>2</sup> or 170 thousand inhabitants (London Air, 2018; Association for the

Monitoring of Air Quality in the Île-de-France, 2018; Regional Agency for the Protection of the Environment of Lazio, 2020; Madrid Air Quality Portal, 2022; Berlin Air Quality Monitoring Network, 2019; Ministry of Environment and Energy, 2022). Similarly, the city of Nicosia operates only two AQ monitoring stations, corresponding to ca. 1 station every 50 km<sup>2</sup> or 150 thousand inhabitants (Department of Labour Inspection, 2021). The limited number of fixed air quality monitoring stations may result in missing localized pollution hot spots, preventing them from capturing spatial variabilities in urban areas.

Low-cost AQ sensors have evolved rapidly over the last few decades. Among all types of LCSs of gaseous pollutants, modern electrochemical (EC) sensors typically exhibit a wide detection range, fast enough response, as well as adequate selectivity and sensitivity that can qualify them for AQ measurements (He et al., 2023). Additional advantages such as simple and robust operation, low power consumption and portability, combined with ease of installation, allow their deployment in dense AQ networks (Bulot et al., 2019; Bilek et al., 2021; Frederickson et al., 2022; deSouza et al., 2022; Raheja et al., 2023).

The accuracy of EC sensors when tested under controlled (laboratory) conditions can range within several tens of percent (Collier-Oxandale et al., 2020; Pang et al., 2017; Castell et al., 2017). Similarly, the accuracy of the PM sensor employed in the Vaisala AQT530 is in the order of 5% according to the manufacturer (AQT530, 2023), which is at least one order of magnitude higher compared to the respective values of reference PM instruments; e.g.,  $\pm 0.75\%$  for the tapered element oscillating microbalance (TEOM; TEOM 1405-DF, 2020). When compared against reference instruments under field 50 conditions, the performance of low-cost AQ sensors has been shown to deviate further. This is not surprising considering that LCSs can be strongly influenced by their operating environmental conditions, including air temperature, relative humidity (RH), and the presence of other gaseous pollutants that can induce a measurable signal (Spinelle et al., 2015a, 2015b; Lewis et al., 2016; Borrego et al., 2016; Castell et al., 2017; Cui et al., 2021).

The performance limitations of EC sensors are primarily related to the nature of the redox reactions at the surface of the sensing 55 electrode (Stetter and Li, 2008). When gas molecules adsorb onto the electrode, they undergo oxidation or reduction, producing an electrical current proportional to their concentration. This process, and consequently the accuracy of EC sensors, is significantly influenced by the operating temperature and relative humidity (Wei et al., 2018). Low-cost PM sensors, on the other hand, typically rely on optical methods using light scattered by the sampled particles to infer their concentration, and in some models also their size (Karagulian et al., 2019; Loizidis et al., 2025). They can be divided into two categories: particle 60 photometers and particle counters (McMurry, 2000). Both methods provide signals that are proportional to the PM concentration, as well as on the optical properties of the sampled particles defined by their size, morphology and composition. Considering that information on particle morphology and composition is hard to obtain, we typically assume that all the particles are spherical and have a refractive index (and density when determining their mass) corresponding to polystyrene latex or ammonium sulphate aerosol particles (Marx and Mulholland, 1983). These assumptions contribute to the measurement

uncertainty when sampling ambient atmospheric aerosol particles. When the sampled particles are hygroscopic, they can take up a significant amount of water vapour from the ambient air, consequently increasing their size and altering their refractive index significantly (Carslaw et al., 2022); a phenomenon that provides another source of uncertainty in low-cost PM sensors. As shown by a growing body of literature reports, existing LCSs cannot meet the quality objectives for use in regulatory observations that require at most 15% uncertainty (specifically expressed as relative expanded uncertainty, REU, of short-term 70 (24-hour) mean concentrations defined by EU Directive 2024/2881, 2024) for CO, NO<sub>2</sub>, and O<sub>3</sub>, and 25% for PM measurements. Similarly, they often fail to meet the criteria for indicative measurements, which call for a 25% REU for CO, NO<sub>2</sub>, and O<sub>3</sub>, 35% for PM<sub>2.5</sub> and 50% for PM<sub>10</sub>, as described in the same Directive. However, they have been found useful in other applications, including identification of air pollution hot spots (Gao et al., 2015, Baruah et al., 2023, Feinberg et al., 2019), distinction between local and non-local pollution sources (Heimann et al., 2015, Popoola et al., 2018), and high- 75 resolution AQ mapping (Schneider et al., 2017) among others, which require lower accuracy. In such studies, LCSs have been integrated as part of networks for multi-point AQ measurements, consequently increasing the spatiotemporal resolution of AQ monitoring networks.

The high uncertainties associated with LCS measurements can in principle be reduced by more laborious calibration models than those provided by the manufacturers. These models can be developed through well-designed laboratory tests (e.g., 80 Nagendra et al., 2019) and/or by field measurements whereby the LCSs are collocated with reference instruments over long periods of time (Raheja et al., 2023, Bisignano et al., 2022, Crawford et al. 2021, Kim et al., 2018). Machine-learning calibration methods can additionally be employed to further improve the quality of the produced data (Bisignano et al., 2022; Baruah et al., 2023; Cabaneros et al., 2019). Such algorithms are already embedded in some AQ monitors that employ low-cost gas and PM sensors (e.g., the Vaisala AQT530; Petäjä et al., 2021). Despite these efforts, critical questions still need to 85 be addressed, including the ability of LCSs to capture spatial and temporal variations of pollutants, and their effectiveness in identifying air pollution hotspots.

The Vaisala Air Quality Transmitter (AQT530) is one of the commercially available cost-effective air quality monitors that incorporates proprietary algorithms for compensating effects related to variable environmental conditions and sensor aging (AQT530; 2023). Comprehensive laboratory tests of the AQT530 at the Air Quality Sensor Performance Evaluation Center 90 (AQ-SPEC) of the South Coast Air Quality Management District in California, indicated that the measurements reported by its CO LCS exhibited high accuracy (i.e., > 91%) and correlated well (i.e., R<sup>2</sup> > 0.95) with measurements from reference instruments (AQ-SPEC, 2022a). At field conditions, the accuracy of the CO LCS remained satisfactory (i.e., > 78%), while the measurements it reported still correlated well (i.e., R<sup>2</sup> > 0.95) with those from reference instruments when averaged over 1 hour (AQ-SPEC, 2022b). Similarly, the O<sub>3</sub> and NO<sub>2</sub> LCSs of the AQT530 exhibited variable accuracies (i.e., from ca. 60 to 95% 95) and high correlations (i.e., R<sup>2</sup> > 0.90), depending on the laboratory concentrations, temperature and RH (AQ-SPEC; 2022a) tested. Field evaluations, however, revealed a marked decline in the accuracy of these sensors, overestimating the actual gas concentrations by 20 and 76% (AQ-SPEC, 2022a) and weak correlations with reference instruments (R<sup>2</sup> values ranging from 0.15 to 0.62; AQ-SPEC; 2022b). This discrepancy between laboratory and field performance highlights the importance

of field evaluations of LCSs, warranting extensive location-specific tests before deployment, especially when the expected environmental conditions are highly variable.

Here we go a step further from assessing the performance of the Vaisala AQT530 monitors, evaluating their ability to capture spatial and temporal differences of pollutants within a city agglomerate that is characterized by strong diurnal temperature, RH and pollutant concentration variations. To achieve that, we carried out measurements over a period of 19 months with two monitors collocated with reference instruments at two AQ monitoring stations (a traffic and an urban background station) in  
the city of Nicosia, Cyprus. The data we collected allowed us to determine whether the spatiotemporal pollutant differences reported by AQT530 monitors are real or not.

## 2 Methods

### 2.1 Description of Air-Quality Measurement Stations

Figure 1 shows an aerial photograph of part of the city of Nicosia, with the locations of the two AQ measurement stations. The  
distance between the two stations is 3.5 km. The Nicosia Traffic Station (referred to as TRS from now on), is ten metres from one of the main and busiest city avenues (ef:see Fig. 1), which is typically congested during the morning and the afternoon rush hours. The site is operated by the Department of Labour Inspection (DLI), which is part of the Ministry of Labour and Social Insurance of Cyprus, and one of the two reference AQ stations of Nicosia (Department of Labour Inspection, 2021). Measurements at the station are conducted according to guidelines described in the relevant EC Directives and the  
corresponding national laws defining the specifications that the employed instruments have to meet, as well as the procedures that must be followed for operating and maintaining them (see Directive 2024/2881, 2024 for more details).

The Nicosia Cyprus Atmospheric Observatory (CAO) station is located at the campus of the Cyprus Institute (CyI) next to the Athalassa forestry park in Nicosia, and can be considered an urban background station (referred to as UBS from now on; ef:see Fig. 1). The area around the UBS station is sparsely populated with no significant local pollution sources in its vicinity (i.e.,  
low traffic density, industry, commercial centres, restaurants, etc.). The site is sporadically influenced by minor local traffic due to the trans-pass of a small number of vehicles through the CyI campus. The concentrations of different gaseous pollutants and PM are continuously monitored at this station for research purposes.

**Figure 1: Aerial photograph showing the locations of the Nicosia Traffic Station (35.1520N, 33.3479E) and of the Nicosia CAO Station (35.1460N, 33.3806E), where the AQT530 monitors were installed during the study period (©Google Earth 2023). Map lines delineate study areas and do not necessarily depict accepted national boundaries.**

## 2.2 Instrumentation

The reference instruments used at the two stations are standard analysers for measuring the concentration of gaseous pollutants, and the TEOM Model 1405-DF for the PM measurements (ef.see Table 1). The specifications of the instruments used at the

TRS and UBS are provided in Tables S1 and S2 of the supplement, respectively.

At the TRS, zero and span checks are performed daily using station gas and zero-air generator to monitor analyser drifts. All reference gas analysers are calibrated monthly using high-concentration certified transfer gas, in accordance with manufacturer guidelines and EN standards. O<sub>3</sub> analysers are calibrated every three months at the National Reference Laboratory while O<sub>3</sub> span and zero checks are carried out daily. The measurements are validated and reported by DLI at a one-hour time resolution.

At the UBS, span and zero checks are performed weekly for all reference gas analysers using certified high-concentration gas cylinders and zero-air generator, in accordance with guidelines provided by the manufacturers and European (EN) standards. Calibration of gas analysers is performed monthly. The O<sub>3</sub> analysers are calibrated every three months at the National Reference Laboratory. O<sub>3</sub> span and zero checks are performed daily. Gas concentrations from the reference analysers, expressed in ppb, are reported at one-hour time resolution.

**Table 1: Instruments operated at the Nicosia traffic (TRS) and CAO (UBS) air quality monitoring stations that provide the reference measurements used in this study. The limit of detection (LoD) of each instrument is also provided.**

|                 | Model                 |                      | LoD                   |                                         |
|-----------------|-----------------------|----------------------|-----------------------|-----------------------------------------|
|                 | TRS                   | UBS                  | TRS                   | UBS                                     |
| CO              | Ecotech Serinus 30    | Teledyne Model T300  | 40 ppb                | < 40 ppb                                |
| NO <sub>x</sub> | Ecotech Serinus 40    | Teledyne Model T500U | 0.4 ppb               | < 0.04 ppb                              |
| O <sub>3</sub>  | Thermo Scientific 49i | Teledyne Model T400  | 0.5 ppb               | < 0.4 ppb with 80 Sample Digital Filter |
| PM              | TEOM Model: 1405-DF   | TEOM Model: 1405-DF  | < 5 µg/m <sup>3</sup> | < 5 µg/m <sup>3</sup>                   |

Two Vaisala Air Quality Transmitter-Monitors (AQT530) series and Weather Transmitters (WXT530) series are employed in this study. These monitors include four Alphasense B-series EC sensors for trace gas measurements (i.e., CO, NO<sub>2</sub>, NO, 145 and O<sub>3</sub>), and a particle counter (Model LPC200) that measures aerosol mass concentrations in two size fractions (i.e., mass concentrations of particles smaller than 2.5 and 10 µm; PM<sub>2.5</sub> and PM<sub>10</sub>, respectively). The specifications of the sensors as those are reported by the manufacturer are provided in Table 2. The AQT530 monitors also have a built-in Vaisala HUMICAP® humidity and temperature probe (Model HMP110). The WXT530 transmitter provides measurements of the wind direction and speed, rainfall, temperature, and absolute pressure.

The signals from the gas sensors (reported in mV) used in the AQT530 monitors are converted to concentrations (in ppb) using proprietary calibration algorithms developed by Vaisala, which differ from those provided by Alphasense, compensating for the impact of ambient conditions and aging of the sensor elements. We should note here that during the course of the measurements, firmware updates that included new calibration models for the NO<sub>2</sub> and O<sub>3</sub> LCSs became available by Vaisala. More specifically, the AQT530 monitors were updated from firmware 3.4 to 3.5 on 25 August 2022, in order to improve the 155 measurements reported by the NO<sub>2</sub> sensors, and to firmware 3.6 on 26 January 2023, to improve the measurements reported by the O<sub>3</sub> sensors. Comparison of the measurements reported by the LCSs using the two firmware is discussed further in sections 2.3 and 3.5.

Prior installation at the two stations, the AQT530 monitors were collocated at the TRS station for one week (from 5 to 11 November 2021) for inter-comparison, testing differences between the sensors while measuring the same concentrations of 160 gaseous pollutants and particles. Both AQT530 monitors were placed outdoors at a distance of a few cm from each other and 1 m from the inlet of the reference instrumentation. Following this period, one monitor was relocated to the UBS station, while the other remained at the TRS station. Both Vaisala AQT530 monitors, referred to as VSL<sub>TRS</sub> (at TRS) and VSL<sub>UBS</sub> (at UBS), collected data from 2 December 2021 to 22 June 2023. Over this 19-month period, the sensors provided continuous time series of measurements, with only minor interruptions.

**Table 2: Specifications and characteristics determined by field tests of the LCSs used in the AQT530 monitor for gaseous pollutants (NO<sub>2</sub>, NO, O<sub>3</sub>, and CO) and particles (PM<sub>2.5</sub>, PM<sub>10</sub>). Published by Vaisala | B211817EN-F © Vaisala 2023 (AQT530; 2023).**

| Specification <sup>1)</sup>                                | NO <sub>2</sub> | NO       | O <sub>3</sub> | CO         | PM <sub>2.5</sub>      | PM <sub>10</sub>       |
|------------------------------------------------------------|-----------------|----------|----------------|------------|------------------------|------------------------|
| Highest concentration limit                                | 2000 ppb        | 2000 ppb | 2000 ppb       | 10,000 ppb | 1000 µg/m <sup>3</sup> | 2500 µg/m <sup>3</sup> |
| Size range                                                 | –               | –        | –              | –          | 0.6–2.5 µm             | 0.6–10 µm              |
| Detection limit                                            | 5 ppb           | 5 ppb    | 5 ppb          | 10 ppb     | 0.1 µg/m <sup>3</sup>  | 0.1 µg/m <sup>3</sup>  |
| Correlation with reference (R <sup>2</sup> ) <sup>2)</sup> | 0.80            | 0.75     | 0.80           | 0.85       | 0.65                   | 0.75                   |
| Accuracy <sup>3)</sup>                                     | 5 ppb           | 15 ppb   | 6 ppb          | 183 ppb    | 9 µg/m <sup>3</sup>    | 13 µg/m <sup>3</sup>   |
| Unit-to-unit correlation (R <sup>2</sup> ) <sup>4)</sup>   | 0.98            | 0.96     | 0.95           | 0.97       | 0.97                   | 0.97                   |
| Precision                                                  | 3 ppb           | 3 ppb    | 4 ppb          | 25 ppb     | 2 µg/m <sup>3</sup>    | 3 µg/m <sup>3</sup>    |

<sup>1)</sup>All values are based on 1-hour averages using only the factory calibration. Values are obtained from field tests carried out in major climate zones against reference instruments. The values represent typical values and may be different based on the location.

<sup>2)</sup>R<sup>2</sup> values determined when correlating the measurements with reference grade instrument derived from all field tests.

<sup>3)</sup>Mean absolute error determined by comparing the LCS measurements with reference measurements.

<sup>4)</sup>Mean absolute difference determined by subtracting the instantaneous readings of the AQT530 monitors LCSs from their mean when the concentration of the gases was maintained constant during laboratory tests.

### 2.3 Data processing and analysis

Negative values reported by all sensors in the AQT530 monitors were flagged and removed as suggested by the manufacturer. All measurements were then averaged over a period of an hour for comparison with the reference measurements. To determine the impact of temperature and RH variabilities on the performance of the sensors we divided the whole dataset into different temperature (i.e., < 10 °C, 10–20 °C, 20–30 °C and > 30 °C) and RH (i.e. < 30%, 30–55%, 55–75% and > 75%) ranges, following the same procedure described by Papaconstantinou et al. (2023).

The performance of the sensors was evaluated by directly correlating and comparing the reported concentrations with measurements by the respective reference instruments to determine the associated errors at each sampling station. The parameters used to do so were the coefficient of determination (R<sup>2</sup>), the Mean Bias Error (MBE), the Mean Relative Error (MRE) and the Mean Absolute Error (MAE), defined respectively as follows:

$$R^2 = 1 - \frac{\sum_{i=1}^N (C_{V,i} - C_{ref,i})^2}{\sum_{i=1}^N (C_{V,i} - \bar{C}_{ref})^2}, \quad (1)$$

$$MRE = \frac{1}{N} \sum_{i=1}^N \left( \frac{|C_{V,i} - C_{ref,i}|}{C_{ref,i}} \right) \times 100, \quad (2)$$

$$MBE = \frac{1}{N} \sum_{i=1}^N C_{V,i} - C_{ref,i}, \text{ and} \quad (3)$$

$$MAE = \frac{1}{N} \sum_{i=1}^N |C_{V,i} - C_{ref,i}|. \quad (4)$$

In all the equations listed above, N is the total number of data points, whereas C<sub>V,i</sub> and C<sub>ref,i</sub> are the concentrations (expressed in ppb) measured respectively by the sensors employed in the AQT530 monitors and the reference instruments at time i.

To investigate whether the differences between LCS measurements at the two different stations, or between the LCS and the reference measurements at the same station are statistically significant, we used the Wilcoxon rank-sum (WRS) test (see details in section S.4 in the Supplement). Differences that are not statistically significant (i.e., when  $p > 0.05$ ) indicate that the data are samples from continuous distributions with equal medians.

The measurements from the NO<sub>2</sub> and O<sub>3</sub> LCSs were also divided into two periods, corresponding to the upgrades of their firmware. The performance of the sensors for each period (namely before and after their firmware update) is assessed with target diagrams, provided in section 3.5, where the vector distance from their origin shows the level of bias and variance of each sensor against reference measurements. The vector is the root mean square error (RMSE) calculated as:

$$\left(\frac{RMSE}{\sigma_{ref}}\right)^2 = \left(\frac{MBE}{\sigma_{ref}}\right)^2 + \left(\frac{CRMSE}{\sigma_{ref}}\right)^2, \quad (5)$$

where CRMSE is the centred root mean square error, which is the RMSE corrected for bias, and MBE is the mean bias error.

All parameters were normalized by the standard deviation of the reference measurements,  $\sigma_{ref}$ . The horizontal line passing through the centre of the target diagram corresponds to zero bias, with the points above or below it corresponding respectively to overestimations or underestimations compared to the reference measurements. When the standard deviation of the LCS measurements is higher or lower compared to those reported by the reference instruments, the points fall on the right or left quadrant of the target diagram, respectively.

### 205 3 Results and Discussion

#### 3.1 Collocated measurements and overall performance of Vaisala AQTs

Table 2Table 3 shows the results from the one-week period we collocated the LCSs with reference instruments at the TRS where we tested the AQT530 monitors against each other (unit-to-unit comparison) and against the reference instruments. The highest unit-to-unit agreements (i.e., MRE less than ca. 10%) and correlations ( $R^2$  values greater than 0.99) were observed for CO, NO and PM<sub>10</sub>. The rest of the LCS (i.e., NO<sub>2</sub>, O<sub>3</sub> and PM<sub>2.5</sub>) exhibit good to moderate unit-to-unit agreements (MRE up to ca. 50%), and inter-correlations with  $R^2$  values ranging from 0.72 to 0.93. Correlation plots, including linear fits between the measurements reported by the sensors of the two AQT530 monitors are provided in Fig. 2.

When compared with reference measurements, the CO sensors in the two AQT530 monitors exhibit good agreement (MAE < 175 ppb) and the highest correlation ( $R^2 > 0.96$ ), followed by the PM<sub>10</sub>, NO<sub>2</sub>, NO, O<sub>3</sub> and PM<sub>2.5</sub> sensors, which show higher levels of uncertainty and error in comparison to the reference measurements as summarised in Table 2Table 3 (ef-see Fig. S2 for the respective correlation plots).

To investigate whether the unit-to-unit differences during the collocation week are statistically significant, we used the WRS test. This allowed us to further evaluate if the sensors produce comparable/reproducible results that can be used for the scope of this study. The results of these test show that the unit-to-unit differences of the NO, O<sub>3</sub>, and PM<sub>2.5</sub> sensors were statistically significant, indicating that the agreement of the sensors is not adequate for determining spatial differences. In contrast, the

respective differences of the CO, NO<sub>2</sub>, and PM<sub>10</sub> measurements were not statistically significant. Among those, the CO and PM<sub>10</sub> measurements exhibit the highest inter-correlation and agreement, with R<sup>2</sup> and slopes of the respective fitted lines that almost unity (ef.see Fig. 2). Considering that, only the CO and PM<sub>10</sub> sensors were tested for their ability to capture spatial differences between different stations (ef.see section 3.2 below).


Figure 2: Correlation between hourly-averaged measurements from the LCSs in the two Vaisala AQT530 monitors over a period of one week that those were collocated at TRS. The y-axis (VSL<sub>TRS</sub>) corresponds to the measurements with the AQT530 monitor operated at TRS after the collocation period, whereas the x-axis (VSL<sub>UBS</sub>) to the measurements of the monitor that moved to UBS after the collocation week. The slope and intercept of the linear regression fitting ( $y = ax + b$ ) for each sensor pair is also indicated. The black dashed lines correspond to the 1:1 correlation.

Table 32: Correlation and mean absolute differences between the measurements reported by the LCSs in the two AQT530 monitors and between each LCS with the respective reference instrument during the period we collocated them with reference instruments at the TRS.

|  | Correlation between LCS measurements ( $R^2$ ) | Mean Absolute Difference between LCS | Median $\pm$ standard deviation | Median $\pm$ standard deviation of LCS residuals | Correlation between LCS & reference measurements ( $R^2$ ) | Mean Absolute Error between LCS & reference measurements |
|--|------------------------------------------------|--------------------------------------|---------------------------------|--------------------------------------------------|------------------------------------------------------------|----------------------------------------------------------|
|  |                                                |                                      |                                 |                                                  |                                                            |                                                          |

|                         |      | measurements           |                    |                    | (VSL <sub>TRS</sub> - VSL <sub>UBS</sub> ) |                    |                    |                         |                         |
|-------------------------|------|------------------------|--------------------|--------------------|--------------------------------------------|--------------------|--------------------|-------------------------|-------------------------|
|                         |      |                        | VSL <sub>TRS</sub> | VSL <sub>UBS</sub> |                                            | VSL <sub>TRS</sub> | VSL <sub>UBS</sub> | VSL <sub>TRS</sub>      | VSL <sub>UBS</sub>      |
| <b>CO</b>               | 1.00 | 36.06 ppb              | 239.25±393.62      | 238.79±388.54      | 32.79±17.90                                | 0.96               | 0.97               | 143.27 ppb              | 172.68 ppb              |
| <b>NO<sub>2</sub></b>   | 0.72 | 11.55 ppb              | 25.83±15.70        | 26.83±17.03        | -1.50±9.10                                 | 0.66               | 0.70               | 13.48 ppb               | 9.60 ppb                |
| <b>NO</b>               | 1.00 | 22.11 ppb              | 93.33±54.20        | 103.37±65.43       | -10.25±5.34                                | 0.60               | 0.70               | 67.48 ppb               | 75.86 ppb               |
| <b>O<sub>3</sub></b>    | 0.83 | 12.28 ppb              | 11.50±13.10        | 5.67±12.30         | 2.00±5.32                                  | 0.51               | 0.63               | 8.38 ppb                | 8.18 ppb                |
| <b>PM<sub>2.5</sub></b> | 0.93 | 2.82 µg/m <sup>3</sup> | 5.72±3.36          | 8.12±4.81          | -2.34±1.70                                 | 0.44               | 0.61               | 12.70 µg/m <sup>3</sup> | 9.98 µg/m <sup>3</sup>  |
| <b>PM<sub>10</sub></b>  | 0.99 | 2.46 µg/m <sup>3</sup> | 26.56±27.17        | 28.68±26.05        | -1.60±2.78                                 | 0.67               | 0.71               | 19.78 µg/m <sup>3</sup> | 18.11 µg/m <sup>3</sup> |

Following the collocated measurements described above, one of the AQT530 monitors was installed at the UBS while the other remained at the TRS, and both systems were allowed to collect data over a period of 19 months, as described in section 2.2. Figure 3 shows the correlation between the measurements recorded by the monitor employed at the TRS (Fig. 3a-f) and UBS (Fig. 3g-l) against the respective measurements by the reference instruments. All the data points are colour coded based on the measurement season, with blue dots corresponding to the cold and red to the warm seasons (ef.see Fig. S2 and S3 in the 240 supplement that provide the same data in the form of time series). Statistical measures of the differences between the VSL<sub>TRS</sub> and the VSL<sub>UBS</sub> measurements against their respective reference measurements are provided in Table 3Table 4. Overall, the CO measurements from both AQT530 monitors (ef.see Fig. 3a for VSL<sub>TRS</sub>, and Fig. 3g for VSL<sub>UBS</sub>) exhibit good 245 agreement with the respective reference measurements as reflected by the relatively low MRE values (-50.51% for VSL<sub>TRS</sub>, and 38.32% for VSL<sub>UBS</sub>), and the high correlation with those reported by the respective reference instrument, exhibiting R<sup>2</sup> values of 0.91 for VSL<sub>TRS</sub>, and 0.70 for VSL<sub>UBS</sub>. The O<sub>3</sub> LCS measurements (ef.see Fig. 3d and 3j) also exhibit deviations 250 from their respective reference measurements (MRE is 89.22% for VSL<sub>TRS</sub> and -7.13% for VSL<sub>UBS</sub>), but weaker correlations compared to the CO LCSs (R<sup>2</sup> values for VSL<sub>TRS</sub> and VSL<sub>UBS</sub> are 0.18 and 0.16, respectively). The performance of the NO<sub>2</sub> sensors (Fig. 3b and 3h) was among the poorest as indicated by the high MREs (224.48% for VSL<sub>TRS</sub> and 732.98% for VSL<sub>UBS</sub>) and the lack of correlation (R<sup>2</sup> value of 0.012 for VSL<sub>TRS</sub> and 0.0014 for VSL<sub>UBS</sub>). Similarly, the NO sensors (Fig. 3c and 3i) 255 exhibit very high MREs (1.17×10<sup>4</sup>% for VSL<sub>TRS</sub> and 1.03×10<sup>5</sup>% for VSL<sub>UBS</sub>), mainly overestimating the reference concentrations, and extremely low correlation (R<sup>2</sup> being in the range of 10<sup>-2</sup> and 10<sup>-4</sup> for VSL<sub>TRS</sub> and VSL<sub>UBS</sub>, respectively). In general, all gas sensors show weaker correlations during the warm period between June and September, corroborating previous findings reported by our group (Papacostantinou et al., 2023). Our results showed lower yet comparable R<sup>2</sup> values for the CO and the O<sub>3</sub> sensors, but significantly lower for the NO and NO<sub>2</sub> measurements compared to the respective values reported in the AQ-SPEC field evaluation study (AQ-SPEC; 2022b). This discrepancy can be attributed to the broader range

of pollutant concentrations and environmental conditions (i.e., temperature and RH) encountered during the longer deployment period in our study: 19 months compared to the 3 months of the AQ-SPEC study.

Apart from the relative errors and correlations discussed above, it is important to investigate whether the month-to-month trends captured by the LCS measurements are similar to those from the reference instruments. Overall, the CO and PM sensors 260 captured the month-to-month trend over the entire period of the measurements similarly to the reference instruments. In contrast, however, the measurements by the O<sub>3</sub> and NO LCSs, as well as part of those by the NO<sub>2</sub> LCSs (measurements corresponding to the first half of the study period), exhibited different overall trends and significant deviations against reference measurements, particularly during the summer periods (ef.see Fig. S2 in the supplement). This difference in the temporal trends indicates that the performance of these sensors is affected more strongly by the high temperature and RH conditions 265 compared to the CO LCSs, warranting further investigation and efforts to improve their performance.

The PM concentrations reported by the VSL<sub>TRS</sub> and VSL<sub>UBS</sub> AQT530 monitors are lower than the reference measurements, as reflected by the negative MRE values shown in Table 3Table 4. Compared to PM<sub>10</sub>, the PM<sub>2.5</sub> measurements reported by the AQT530 monitors (Fig. 3e and 3k) exhibit higher deviations against those provided by the reference instruments in both stations (Fig. 3f and 3l). This is because the PM LCSs have a high cut-off diameter (50% detection efficiency for particle sizes 270 of 0.6  $\mu\text{m}$ ; Vaisala, 2022), resulting in a portion of particles going undetected. These undetected particles contribute proportionally more to the PM<sub>2.5</sub> than to the PM<sub>10</sub> mass concentration. Overall, the PM concentrations reported by the VSL<sub>TRS</sub> monitor show greater discrepancies compared to those from the VSL<sub>UBS</sub> monitor when measured against the respective reference values (ef.see Table 3Table 4). Despite that only the PM<sub>10</sub> measurements showed good unit-to-unit reproducibility when the two AQT530 monitors were collocated at the TRS in the beginning of our study period (ef.see Table 2Table 3), the 275 difference in the PM measurements between the two stations, as observed here, can be attributed to the higher fraction of particles smaller than the cut-off diameter of the PM sensors at the TRS compared to the UBS (see Fig. S6 in the supplement). This is highly possible considering that freshly emitted and smaller particles, which are characteristic of traffic emissions (Zhu, 2002), should be higher at the TRS.

We should highlight here that the agreement between PM LCSs and reference instruments improved significantly during 280 periods with dust events; a phenomenon that is more frequent and intense during spring and autumn in the region (Yukhymchuk et al. 2022, Kezoudi et al. 2021). More specifically, the MRE between the PM<sub>2.5</sub> and PM<sub>10</sub> measurements reported by the LCSs decreased respectively from -74.16 to -59.98% and from -64.26 to -55.48% at the TRS, and from -57.28 to -27.75% and from -43.98 to -28.78% at the UBS. Similarly, the R<sup>2</sup> values of PM<sub>2.5</sub> measurements increased from 0.12 during the non-dust period (i.e., summer and winter) to 0.43 during the dust period (i.e., spring and autumn) for VSL<sub>TRS</sub>, and from 0.18 to 0.47 for VSL<sub>UBS</sub>, 285 respectively (see Fig. S4 in the supplement). The R<sup>2</sup> values for PM<sub>10</sub> measurements increased from 0.32 to 0.78 for VSL<sub>TRS</sub> and from 0.47 to 0.92 for VSL<sub>UBS</sub>, respectively (ef.see Fig. S5 in the supplement). Another possible reason contributing to the improved performance of the PM sensors during the dust events is the similarity in the optical properties of the micron-sized dust particles to those of calibration aerosol particles (AQT530, 2023).

TRS

UBS

**Figure 3: Correlation between hourly averaged measurements recorded by the VSL<sub>TRS</sub> (Fig.3-a-f, left panel) or the VSL<sub>UBS</sub> (Fig. 3-g-l, right panel) AQT530 monitors, and those provided by the respective reference instruments. The black dashed lines correspond to the 1:1 correlation. The data points are colour-tagged with respect to season, with blue indicating the cold and red the warm seasons.**

**Table 43:** Summary statistics, including the number of useful measurements (N) recorded by the Vaisala AQT530 monitors, and the mean value of hourly averaged concentrations measured by the respective LCSs and the reference instruments for the entire study period, together with the Standard Deviation (SD) for each case, as well as the associated values of the MRE, MBE, MAE and R<sup>2</sup>.

| Pollutant                                  | Vaisala N                | Vaisala Mean $\pm$ SD | Reference Mean $\pm$ SD | MRE (%)                       | MBE     | MAE    | R <sup>2</sup>                 | Vaisala LoD           |
|--------------------------------------------|--------------------------|-----------------------|-------------------------|-------------------------------|---------|--------|--------------------------------|-----------------------|
| <b>CO (ppb)</b>                            | VSL <sub>TRS</sub> 13216 | 190.16 $\pm$ 226.41   | 357.89 $\pm$ 292.26     | -50.51                        | -165.44 | 169.47 | 0.91                           | 10 ppb                |
|                                            | VSL <sub>UBS</sub> 9963  | 129.84 $\pm$ 136.50   | 235.53 $\pm$ 169.25     | 38.32                         | -66.44  | 85.35  | 0.70                           |                       |
| <b>NO<sub>2</sub> (ppb)</b>                | VSL <sub>TRS</sub> 13216 | 32.35 $\pm$ 29.23     | 14.82 $\pm$ 9.80        | 224.48                        | 17.16   | 18.57  | 0.012                          | 5 ppb                 |
|                                            | VSL <sub>UBS</sub> 12325 | 27.24 $\pm$ 24.47     | 8.07 $\pm$ 9.01         | 732.98                        | 18.87   | 19.46  | 0.0014                         |                       |
| <b>NO (ppb)</b>                            | VSL <sub>TRS</sub> 13216 | 111.05 $\pm$ 102.63   | 11.72 $\pm$ 22.71       | 1.17 $\times$ 10 <sup>4</sup> | 98.33   | 98.62  | 5.64 $\times$ 10 <sup>-4</sup> | 5 ppb                 |
|                                            | VSL <sub>UBS</sub> 12354 | 86.87 $\pm$ 84.88     | 3.30 $\pm$ 11.50        | 1.03 $\times$ 10 <sup>5</sup> | 60.35   | 60.43  | 0.0064                         |                       |
| <b>O<sub>3</sub> (ppb)</b>                 | VSL <sub>TRS</sub> 13216 | 40.69 $\pm$ 47.20     | 28.34 $\pm$ 16.85       | 89.22                         | 11.94   | 23.53  | 0.18                           | 5 ppb                 |
|                                            | VSL <sub>UBS</sub> 10453 | 31.23 $\pm$ 26.06     | 36.18 $\pm$ 16.20       | -7.13                         | -5.19   | 15.54  | 0.16                           |                       |
| <b>PM<sub>2.5</sub> (µg/m<sup>3</sup>)</b> | VSL <sub>TRS</sub> 13216 | 5.43 $\pm$ 6.34       | 19.17 $\pm$ 12.32       | -62.71                        | -13.43  | 13.66  | 0.15                           | 0.1 µg/m <sup>3</sup> |
|                                            | VSL <sub>UBS</sub> 13258 | 7.05 $\pm$ 8.87       | 13.29 $\pm$ 9.07        | -36.84                        | -5.64   | 7.47   | 0.22                           |                       |
| <b>PM<sub>10</sub></b>                     | VSL <sub>TRS</sub> 13216 | 15.62 $\pm$ 19.86     | 38.78 $\pm$ 26.20       | -58.99                        | -22.65  | 23.24  | 0.44                           | 0.1 µg/m <sup>3</sup> |

| ( $\mu\text{g}/\text{m}^3$ ) | VSL <sub>UBS</sub> | 13258 | 18.05 $\pm$ 27.99 | 25.82 $\pm$ 20.10 | -29.51 | -6.75 | 10.54 | 0.56 |
|------------------------------|--------------------|-------|-------------------|-------------------|--------|-------|-------|------|
|------------------------------|--------------------|-------|-------------------|-------------------|--------|-------|-------|------|

As shown in Fig. 3, the measurements of the low-cost gas sensors are in better agreement with those of the reference instruments during the cold periods (i.e., from September to May when RH is higher than 55%) than during the warm periods (i.e., from June to August when RH is decreased below 55%); ef-see Fig. S76 and S98 for the dependence of the LCSs on 300 temperature and Fig. S110 and S132 on the RH. Some of the LCSs (particularly the NO<sub>2</sub>, NO and O<sub>3</sub> sensors) provide measurements that deviate substantially from those of the reference instruments over the warm periods. We have observed similar behaviour in previous measurements with LCSs, and have attributed it to the evaporation of water in the electrolyte of the sensors (Papacostantinou et al., 2023; Alphasense AAN106). We should note here that gas sensors used in the Vaisala AQT530 monitors exhibit better performance compared to the sensors tested in our previous work, which, in contrast to this 305 study, did not fully recover after being exposed to moderate temperature and RH conditions. Although the sensors were of the same model and manufacturer in both studies, this difference can be attributed to the different production batches or the type/design of electrolyte enclosure used.

The measurements of the PM LCSs are in better agreement with those of reference instruments at RH conditions below 30% 310 (ef-see Fig. S124 and S143 in the supplement for VSL<sub>TRS</sub> and VSL<sub>UBS</sub>, respectively). This is because the size and refractive index of the particles change due to water uptake at higher RH conditions, increasing significantly their scattering efficiency, and causing deviations from the reference measurements that are carried out at dry conditions (Titos et al., 2016; Bezantakos et al., 2021). Since typically low RH conditions are associated with high temperatures, PM LCS measurements tend to correlate better with reference measurements at higher temperatures. Agreement between the Vaisala PM<sub>10</sub> with the reference 315 measurements at both stations is better compared to that of PM<sub>2.5</sub> for all temperature and RH conditions due to the reasons described above.

### 3.2 Spatial variabilities determined by the reference instruments and the LCSs

The results shown in Table 3Table 4 indicate that the reference concentrations (averaged over the entire study period) of all the pollutants at the two stations are different, with deviations ranging from ca. 25 to 110%. For example, the mean reference CO concentration at the TRS is 357.9 ppb while the respective value at the UBS is 235.5 ppb. To investigate whether these 320 differences are statistically significant both when the values are averaged over the entire measurement period and on a monthly basis, we used the WRS test. The results of this test show that the differences for all pollutants measured by the reference instruments are significant at a 95% confidence interval for the average values over the entire measurement period and for each individual month (ef-see Tables S3-S56 in the supplement for more details). These differences are expected, as we are comparing two different types of stations: a traffic and an urban background station, which naturally experience varying 325 pollutant levels due to their differing environments.

The same statistical test was used for the measurements by the CO and PM<sub>10</sub> LCSs of the AQT530 monitors operated at the two stations, which were the ones that exhibited high sensor-to-sensor reproducibility during the collocation period as

discussed in section 3.1. For the entire study period (i.e., all 19 months), the mean concentrations determined by the LCSs in the two stations are statistically different at the same confidence interval (i.e., 95%; ef.see Table S3), as also indicated by 330 comparing the respective reference instruments at the two stations. However, when the same test is applied for every individual month, there are a few cases where the LCSs show no statistically significant differences between the two stations while the reference instruments do (see orange-coloured cells in Tables S4-S5). Figure 4 shows time series of the monthly-average concentrations of the pollutants as those were determined by the measurements from the reference instruments (ef.see solid lines) and the LCSs in the AQT530 monitor (ef.see dashed lines), as well as the concentration differences between reference 335 and LCS measurements. The CO sensors seem to report measurements that follow the overall concentration variability as this is captured by the respective reference instruments. The PM<sub>10</sub> measurements reported by the AQT530 monitors generally follow the trend of the reference measurements. The striking peak observed in April 2022 is due to a strong dust event as discussed in section 3.1; a phenomenon that occurs with higher frequency and intensity during this period of the year in the region (Yukhymchuk et al., 2022, Kezoudi et al., 2021). PM<sub>10</sub> concentrations, however, are generally higher at the TRS 340 compared to the UBS, as indicated by the reference instruments (Fig. 4d3e), which is in contrast to what the LCS measurements indicate (Fig. 4e3h). This discrepancy could be explained by the systematic presence of particles smaller than the detection limit of the low-cost PM sensor (i.e., 600 nm) employed in the AQT530 monitor at the TRS compared to the UBS (see Fig S6 in the supplement). This is a plausible explanation considering that TRS is affected to a higher degree by particles from anthropogenic sources that are typically smaller than the LCS detection threshold. In contrast, background particles, which are 345 more likely to represent the atmospheric aerosol at the UBS, are generally larger and thus the majority (if not all) of those are detected and counted by the low-cost PM sensor in the monitor.

Red-shaded areas in Fig. 4 denote the months when the measurements by the LCSs did not exhibit statistically significant differences while the reference instruments did. The respective p-values calculated by the statistical test are tabulated in the supplement (ef.see Tables S4 and S5). The CO LCSs show no statistically significant differences in 2 out of the 19 months 350 (July and August 2022). During these months, the difference of the CO concentrations between the two stations as determined by the LCSs and the reference instruments was below 4 and 80 ppb, respectively (ef.see Fig. 4b). The PM<sub>10</sub> measurements reported by the LCSs show no statistically significant differences in 7 out of the 19 months (i.e., December 2021, February, March and November 2022 and March, April and May 2023; Fig. 4e). The PM concentration differences between the two stations in these cases, as measured by the LCSs, were less than 4  \$\mu\text{g}/\text{m}^3\$  while those measured by the reference instruments were less than 10  \$\mu\text{g}/\text{m}^3\$ . The PM concentration differences between the two stations in these cases, as measured by the LCSs, were below 3.4  \$\mu\text{g}/\text{m}^3\$  while those measured by the reference instruments were below 6–10  \$\mu\text{g}/\text{m}^3\$ .

The ability of the LCSs to capture spatial variabilities is important for their use for the identification of pollution hot-spots. By identifying whether the LCSs are able to capture these spatial differences, one can in principle determine if they can be reliably used for spatial hot-spot recognition. With the exemption of the two warm months, the CO LCSs can be used for capturing the 360 reference month-to-month trends/variabilities, urban gradients and pollution hot spots.

Commented [RP1]: Updated with legends

Figure 4: Time series showing the monthly-average concentrations of CO (a-b) and  $\text{PM}_{10}$  (d-e) measured by the reference instruments (solid lines) and the low-cost gas sensors employed in the Vaisala AQT530 monitors (dashed lines) at the two stations, as well as the differences in the concentrations between the two stations as those were captured by the reference (solid lines) and LCS (dashed lines) measurements for each pollutant (c and f). The blue and orange lines correspond to measurements at TRS and UBS stations, respectively. The red-shaded areas indicate that the monthly differences between the same LCSs employed in the Vaisala monitors for every month are not statistically significant ( $p > 0.05$ ), in contrast to the reference measurements that showed statistically significant different station-to-station concentrations for all the months.

Figure 5 shows the averaged diurnal trends/variabilities on workdays and weekends for CO and  $\text{PM}_{10}$  during the two months 370 of January in our dataset (i.e., for 2022 and 2023) when the mean hourly temperatures ranged from 0.2 to 20.7 °C, and for the two months of June, when the respective values ranged from 15.6 to 37.7 °C; note that the diurnal variations observed in January 2022 and January 2023, as well as in June 2022 and June 2023, exhibited a high degree of similarity (ef.see Fig. S1<sup>54</sup> and S1<sup>65</sup> in the supplement). Tables S6 and S7 in the supplement provide the p-values from the tests using the data shown in Fig. 5. The hourly reference measurements at the two stations are significantly different ( $p < 0.05$ ), with few exemptions 375 mainly for CO during morning and night hours when the concentration difference is insignificant (p values ranging from  $5.3 \times 10^{-02}$  to  $9.5 \times 10^{-01}$ ). The LCSs in the AQTs are able to capture the differences in the diurnal variation between the two stations better during workdays than weekends, mainly because they exhibit higher concentrations (ef.see reference measurements also in Fig. 5) that can be captured with higher fidelity.

More specifically, the CO sensors in the two AQT530 monitors appear to follow better the diurnal variabilities, as these are 380 captured by the reference instruments, during the cold (January 2023 and January 2023) rather than the warm (June 2022 and

June 2023) months. This is not surprising considering that the performance of EC sensors drops at high temperatures and at low concentrations that occur in the warm period (ef.see Fig. 5a-b against 5c-d). What is more, they fail to capture the significant difference observed between reference measurements during the middle of the days in the weekends when the CO concentration differences are small between the two stations (< 80 ppb as indicated by the reference instruments), especially

during the warm period (i.e., between 10.00 am and 6.00 pm in June; ef.see Table S6 in the supplement).

The PM<sub>10</sub> LCSs captured well the diurnal trend/variability of the reference measurements during the weekdays of the cold periods (ef.see Fig. 5e), but fail to do so during the weekends of the same period (ef.see Fig. 5f), most likely due to the lower concentration of particles that are large enough to be detected by these sensors. The PM<sub>10</sub> measurements also fail to capture the trends observed by the respective reference instruments during the warmer months (both during weekdays and weekends;

ef.see Fig. 5g and 5h), due to the same reason. Overall, the ability of the PM<sub>10</sub> measurements by the LCSs to capture the spatial differences was higher in cases where the difference in the concentration at the two stations was higher (i.e., > 10  $\mu\text{g}/\text{m}^3$ ). It should also be noted that these differences were in the opposite direction compared to that when comparing the measurements reported by the reference instruments: i.e., the LCSs concentrations at UBS are systematically higher than those at the TRS, while the opposite is true for the reference measurements. As explained earlier in the same section, this is due to less sub-600

395 nm particles at the UBS compared to TRS station.

**Figure 5: Diurnal variability of hourly-averaged CO (a-d) and PM<sub>10</sub> (e-h) during January (merged data from 2022 and 2023; left column) and June (merged data from 2022 and 2023; right column) as measured by the reference instrument (solid lines) and the LCSs in the Vaisala monitor (dashed lines) at the two stations. The blue and orange lines correspond to measurements at TRS and UBS stations, respectively.**

**Commented [RP2]:** Updated with legends

### 3.3 Temporal variabilities determined by the reference instruments and the LCSs at the same station

The next step was to examine whether the LCSs can capture the temporal variabilities as those are determined by the reference instruments at each station. To do so, we carried out WRS tests between successive months for the reference and the LCS measurements at each of the two stations. Table 4Table 5 shows the calculated p-values for each test, and whether this indicates significant difference ( $p < 0.05$ ). When the reference month-to-month concentration differences are statistically significant, the LCSs can also reproduce this result for most of the cases (indicated by the white cells corresponding either to  $\text{REF}_{\text{TRS}}$ ,  $\text{REF}_{\text{UBS}}$ ,  $\text{VSL}_{\text{TRS}}$ , or  $\text{VSL}_{\text{UBS}}$  in Table 4Table 5). In contrast, when the month-to-month variability of the reference measurements is not statistically significant (blue and orange cells for the measurements at TRS and UBS, respectively), it is more likely for the respective LCSs to suggest the opposite: i.e., that the differences are significant. This can be attributed to the low accuracy of the LCSs compared to the reference instruments.

The CO LCS measurements at the two stations identified accurately the statistically significant differences in ca. 55-60% of the cases (11 and 10 out of the 18 sets of consecutive months at TRS and UBS, respectively). The respective numbers for the  $\text{NO}_2$  and NO LCSs were ca. 60 and 70% for TRS and ca. 70 and 45% for UBS. Regarding  $\text{O}_3$ , the LCSs captured the statistically significant or insignificant variabilities in ca. 65% of the cases at both stations. The  $\text{PM}_{2.5}$  and  $\text{PM}_{10}$  LCSs captured the statistical significance (or non-significance) of the month-to-month variability in ca. 80 and 70%, respectively, at both stations. Considering that the PM sensors, and marginally the NO and  $\text{NO}_2$  sensors, capture accurately the month-to-month variabilities in more than 70% of the cases, they can qualify as appropriate to determine the seasonal variabilities. However, taking into account that the NO and  $\text{NO}_2$  sensors fail to capture the overall trends, reporting values that are significantly different compared to the reference instruments, they can be excluded, leaving only the PM sensors as the most reliable indicator of temporal variabilities.

**Table 54:** P-values determined by the WSR tests between all pairs of consecutive months of measurements using either the reference or the LCS measurements. Blue (for TRS) and orange (for UBS) cells indicate that the month-to-month differences are not statistically significant ( $p > 0.05$ ), whereas white cells indicate the opposite ( $p < 0.05$ ). Colour agreement between the reference and LCS measurements for each pair of consecutive months indicate that the AQT sensors are able to capture the temporal variabilities.

|                 |                           | Dec-Jan                  | Jan-Feb                  | Feb-Mar                  | Mar-Apr                  | Apr-May                  | May-Jun                  | Jun-Jul                  | Jul-Aug                  | Aug-Sep                  | Sep-Oct                  | Oct-Nov                  | Nov-Dec                  | Dec-Jan                  | Jan-Feb                  | Feb-Mar                  | Mar-Apr                  | Apr-May                  | May-Jun                  |
|-----------------|---------------------------|--------------------------|--------------------------|--------------------------|--------------------------|--------------------------|--------------------------|--------------------------|--------------------------|--------------------------|--------------------------|--------------------------|--------------------------|--------------------------|--------------------------|--------------------------|--------------------------|--------------------------|--------------------------|
| CO              | $\text{REF}_{\text{TRS}}$ | 7.0<br>$\times 10^{-01}$ | 2.9<br>$\times 10^{-01}$ | 8.2<br>$\times 10^{-01}$ | 1.7<br>$\times 10^{-01}$ | 1.8<br>$\times 10^{-04}$ | 4.0<br>$\times 10^{-04}$ | 5.2<br>$\times 10^{-04}$ | 7.1<br>$\times 10^{-26}$ | 1.3<br>$\times 10^{-01}$ | 1.8<br>$\times 10^{-01}$ | 7.3<br>$\times 10^{-01}$ | 2.4<br>$\times 10^{-01}$ | 6.1<br>$\times 10^{-02}$ | 4.8<br>$\times 10^{-35}$ | 1.2<br>$\times 10^{-07}$ | 1.5<br>$\times 10^{-02}$ | 3.3<br>$\times 10^{-02}$ |                          |
|                 | $\text{VSL}_{\text{TRS}}$ | 3.0<br>$\times 10^{-01}$ | 7.7<br>$\times 10^{-02}$ | 4.6<br>$\times 10^{-02}$ | 5.5<br>$\times 10^{-01}$ | 3.1<br>$\times 10^{-01}$ | 6.3<br>$\times 10^{-01}$ | 8.3<br>$\times 10^{-01}$ | 2.3<br>$\times 10^{-01}$ | 3.3<br>$\times 10^{-01}$ | 2.9<br>$\times 10^{-01}$ | 7.4<br>$\times 10^{-01}$ | 1.0<br>$\times 10^{-01}$ | 8.8<br>$\times 10^{-01}$ | 2.2<br>$\times 10^{-01}$ | 7.9<br>$\times 10^{-01}$ | 6.7<br>$\times 10^{-02}$ | 5.8<br>$\times 10^{-01}$ |                          |
|                 | $\text{REF}_{\text{UBS}}$ | 2.5<br>$\times 10^{-05}$ | 1.7<br>$\times 10^{-10}$ | 7.4<br>$\times 10^{-24}$ | 2.8<br>$\times 10^{-24}$ | 1.4<br>$\times 10^{-09}$ | 8.2<br>$\times 10^{-09}$ | 4.5<br>$\times 10^{-03}$ | 4.9<br>$\times 10^{-03}$ | 2.4<br>$\times 10^{-17}$ | 2.9<br>$\times 10^{-70}$ | 2.4<br>$\times 10^{-70}$ | 1.3<br>$\times 10^{-01}$ | 1.8<br>$\times 10^{-01}$ | 1.2<br>$\times 10^{-09}$ | 2.2<br>$\times 10^{-09}$ | 3.7<br>$\times 10^{-11}$ | 1.3<br>$\times 10^{-04}$ | 6.3<br>$\times 10^{-06}$ |
|                 | $\text{VSL}_{\text{UBS}}$ | 3.0<br>$\times 10^{-03}$ | 3.2<br>$\times 10^{-01}$ | 5.3<br>$\times 10^{-01}$ | 2.6<br>$\times 10^{-01}$ | 8.2<br>$\times 10^{-01}$ | 3.6<br>$\times 10^{-01}$ | 3.0<br>$\times 10^{-01}$ | 8.2<br>$\times 10^{-01}$ | 2.8<br>$\times 10^{-01}$ | 9.9<br>$\times 10^{-01}$ | 1.0<br>$\times 10^{-01}$ | 4.2<br>$\times 10^{-01}$ | 2.2<br>$\times 10^{-01}$ | 9.4<br>$\times 10^{-01}$ | 1.7<br>$\times 10^{-01}$ | 7.8<br>$\times 10^{-01}$ | 4.4<br>$\times 10^{-01}$ | 1.6<br>$\times 10^{-01}$ |
|                 | $\text{REF}_{\text{TRS}}$ | 1.8<br>$\times 10^{-02}$ | 4.4<br>$\times 10^{-02}$ | 1.6<br>$\times 10^{-04}$ | 5.9<br>$\times 10^{-04}$ | 1.0<br>$\times 10^{+00}$ | 3.2<br>$\times 10^{-04}$ | 6.8<br>$\times 10^{-02}$ | 1.2<br>$\times 10^{-02}$ | 7.7<br>$\times 10^{-02}$ | 2.0<br>$\times 10^{-02}$ | 5.7<br>$\times 10^{-02}$ | 9.0<br>$\times 10^{-02}$ | 1.6<br>$\times 10^{-02}$ | 2.2<br>$\times 10^{-02}$ | 9.9<br>$\times 10^{-02}$ | 2.2<br>$\times 10^{-07}$ | 5.7<br>$\times 10^{-03}$ |                          |
|                 | $\text{VSL}_{\text{TRS}}$ | 6.5<br>$\times 10^{-03}$ | 7.6<br>$\times 10^{-03}$ | 7.4<br>$\times 10^{-02}$ | 5.2<br>$\times 10^{-02}$ | 2.8<br>$\times 10^{-16}$ | 4.7<br>$\times 10^{-16}$ | 1.2<br>$\times 10^{-16}$ | 1.4<br>$\times 10^{-16}$ | 1.6<br>$\times 10^{-16}$ | 2.6<br>$\times 10^{-12}$ | 4.2<br>$\times 10^{-12}$ | 6.1<br>$\times 10^{-12}$ | 7.0<br>$\times 10^{-12}$ | 6.4<br>$\times 10^{-12}$ | 7.2<br>$\times 10^{-12}$ | 4.0<br>$\times 10^{-12}$ | 9.4<br>$\times 10^{-13}$ |                          |
| NO <sub>x</sub> | $\text{REF}_{\text{UBS}}$ | 1.7<br>$\times 10^{-03}$ | 8.8<br>$\times 10^{-01}$ | 1.8<br>$\times 10^{-05}$ | 2.5<br>$\times 10^{-05}$ | 1.2<br>$\times 10^{-02}$ | 4.7<br>$\times 10^{-02}$ | 2.0<br>$\times 10^{-02}$ | 8.6<br>$\times 10^{-02}$ | 9.6<br>$\times 10^{-02}$ | 2.9<br>$\times 10^{-02}$ | 2.0<br>$\times 10^{-02}$ | 5.4<br>$\times 10^{-02}$ | 8.8<br>$\times 10^{-02}$ | 2.5<br>$\times 10^{-02}$ | 2.1<br>$\times 10^{-02}$ | 5.5<br>$\times 10^{-02}$ | 4.0<br>$\times 10^{-02}$ | 2.2<br>$\times 10^{-02}$ |
|                 | $\text{VSL}_{\text{UBS}}$ | 7.6<br>$\times 10^{-04}$ | 6.7<br>$\times 10^{-02}$ | 1.4<br>$\times 10^{-05}$ | 1.7<br>$\times 10^{-15}$ | 2.8<br>$\times 10^{-08}$ | 2.1<br>$\times 10^{-08}$ | 2.0<br>$\times 10^{-08}$ | 1.4<br>$\times 10^{-08}$ | 2.8<br>$\times 10^{-13}$ | 4.9<br>$\times 10^{-09}$ | 3.2<br>$\times 10^{-09}$ | 8.3<br>$\times 10^{-09}$ | 1.1<br>$\times 10^{-09}$ | 7.6<br>$\times 10^{-09}$ | 3.5<br>$\times 10^{-09}$ | 1.4<br>$\times 10^{-09}$ | 1.2<br>$\times 10^{-09}$ |                          |
|                 | $\text{REF}_{\text{TRS}}$ | 1.7<br>$\times 10^{-03}$ | 8.8<br>$\times 10^{-01}$ | 1.8<br>$\times 10^{-05}$ | 2.5<br>$\times 10^{-05}$ | 1.2<br>$\times 10^{-02}$ | 4.7<br>$\times 10^{-02}$ | 2.0<br>$\times 10^{-02}$ | 8.6<br>$\times 10^{-02}$ | 9.6<br>$\times 10^{-02}$ | 2.9<br>$\times 10^{-02}$ | 2.0<br>$\times 10^{-02}$ | 5.4<br>$\times 10^{-02}$ | 8.8<br>$\times 10^{-02}$ | 2.5<br>$\times 10^{-02}$ | 2.1<br>$\times 10^{-02}$ | 5.5<br>$\times 10^{-02}$ | 4.0<br>$\times 10^{-02}$ | 2.2<br>$\times 10^{-02}$ |

|                   |                    |                            |                            |                            |                            |                            |                            |                            |                            |                            |                             |                             |                            |                            |                            |                            |                            |                            |                            |                            |
|-------------------|--------------------|----------------------------|----------------------------|----------------------------|----------------------------|----------------------------|----------------------------|----------------------------|----------------------------|----------------------------|-----------------------------|-----------------------------|----------------------------|----------------------------|----------------------------|----------------------------|----------------------------|----------------------------|----------------------------|----------------------------|
| NO                | REF <sub>TRS</sub> | 7.7<br>× 10 <sup>-01</sup> | 4.5<br>× 10 <sup>-02</sup> | 6.6<br>× 10 <sup>-06</sup> | 1.5<br>× 10 <sup>-25</sup> | 9.4<br>× 10 <sup>-01</sup> | 2.8<br>× 10 <sup>-01</sup> | 2.9<br>× 10 <sup>-02</sup> | 2.8<br>× 10 <sup>-10</sup> | 2.2<br>× 10 <sup>-10</sup> | 1.4<br>× 10 <sup>-06</sup>  | 1.3<br>× 10 <sup>-06</sup>  | 3.3<br>× 10 <sup>-08</sup> | 4.8<br>× 10 <sup>-08</sup> | 6.5<br>× 10 <sup>-01</sup> | 1.1<br>× 10 <sup>-20</sup> | 1.1<br>× 10 <sup>-07</sup> | 3.0<br>× 10 <sup>-02</sup> | 2.3<br>× 10 <sup>-02</sup> |                            |
|                   | VSL <sub>TRS</sub> | 2.1<br>× 10 <sup>-02</sup> | 6.2<br>× 10 <sup>-01</sup> | 3.4<br>× 10 <sup>-05</sup> | 1.2<br>× 10 <sup>-12</sup> | 8.8<br>× 10 <sup>-01</sup> | 4.7<br>× 10 <sup>-10</sup> | 1.4<br>× 10 <sup>-09</sup> | 1.8<br>× 10 <sup>-02</sup> | 6.7<br>× 10 <sup>-12</sup> | 1.1<br>× 10 <sup>-34</sup>  | 1.9<br>× 10 <sup>-34</sup>  | 2.6<br>× 10 <sup>-02</sup> | 6.7<br>× 10 <sup>-12</sup> | 1.1<br>× 10 <sup>-34</sup> | 1.7<br>× 10 <sup>-07</sup> | 4.6<br>× 10 <sup>-08</sup> | 6.1<br>× 10 <sup>-05</sup> | 2.8<br>× 10 <sup>-22</sup> | 3.9<br>× 10 <sup>-14</sup> |
|                   | REF <sub>UBS</sub> | 5.8<br>× 10 <sup>-03</sup> | 1.3<br>× 10 <sup>-01</sup> | 1.3<br>× 10 <sup>-05</sup> | 2.0<br>× 10 <sup>-11</sup> | 5.9<br>× 10 <sup>-01</sup> | 9.9<br>× 10 <sup>-01</sup> | 3.2<br>× 10 <sup>-01</sup> | 2.6<br>× 10 <sup>-01</sup> | 3.6<br>× 10 <sup>-01</sup> | 3.5<br>× 10 <sup>-11</sup>  | 6.8<br>× 10 <sup>-01</sup>  | 8.4<br>× 10 <sup>-01</sup> | 2.2<br>× 10 <sup>-01</sup> | 4.8<br>× 10 <sup>-01</sup> | 5.8<br>× 10 <sup>-02</sup> | 2.3<br>× 10 <sup>-02</sup> | 1.7<br>× 10 <sup>-01</sup> |                            |                            |
|                   | VSL <sub>UBS</sub> | 2.4<br>× 10 <sup>-01</sup> | 1.3<br>× 10 <sup>-01</sup> | 1.1<br>× 10 <sup>-06</sup> | 1.4<br>× 10 <sup>-13</sup> | 7.6<br>× 10 <sup>-07</sup> | 5.7<br>× 10 <sup>-02</sup> | 4.4<br>× 10 <sup>-02</sup> | 1.1<br>× 10 <sup>-03</sup> | 4.3<br>× 10 <sup>-03</sup> | 4.9<br>× 10 <sup>-06</sup>  | 1.4<br>× 10 <sup>-06</sup>  | 1.4<br>× 10 <sup>-06</sup> | 5.2<br>× 10 <sup>-06</sup> | 4.5<br>× 10 <sup>-06</sup> | 2.7<br>× 10 <sup>-06</sup> | 8.1<br>× 10 <sup>-06</sup> | 2.9<br>× 10 <sup>-06</sup> | 1.5<br>× 10 <sup>-06</sup> |                            |
|                   | i                  |                            |                            |                            |                            |                            |                            |                            |                            |                            |                             |                             |                            |                            |                            |                            |                            |                            |                            |                            |
| O <sub>3</sub>    | REF <sub>TRS</sub> | 7.2<br>× 10 <sup>-02</sup> | 1.5<br>× 10 <sup>-05</sup> | 1.1<br>× 10 <sup>-18</sup> | 3.1<br>× 10 <sup>-04</sup> | 6.5<br>× 10 <sup>-01</sup> | 6.4<br>× 10 <sup>-01</sup> | 4.3<br>× 10 <sup>-01</sup> | 4.1<br>× 10 <sup>-22</sup> | 2.5<br>× 10 <sup>-11</sup> | 4.1<br>× 10 <sup>-40</sup>  | 4.3<br>× 10 <sup>-40</sup>  | 6.8<br>× 10 <sup>-06</sup> | 8.1<br>× 10 <sup>-06</sup> | 6.6<br>× 10 <sup>-06</sup> | 3.4<br>× 10 <sup>-06</sup> | 1.8<br>× 10 <sup>-26</sup> | 9.7<br>× 10 <sup>-02</sup> | 4.3<br>× 10 <sup>-02</sup> | 1.0<br>× 10 <sup>-10</sup> |
|                   | VSL <sub>TRS</sub> | 8.4<br>× 10 <sup>-06</sup> | 2.3<br>× 10 <sup>-01</sup> | 3.6<br>× 10 <sup>-29</sup> | 1.7<br>× 10 <sup>-02</sup> | 5.8<br>× 10 <sup>-14</sup> | 1.0<br>× 10 <sup>-21</sup> | 9.2<br>× 10 <sup>-21</sup> | 7.3<br>× 10 <sup>-21</sup> | 4.2<br>× 10 <sup>-21</sup> | 6.2<br>× 10 <sup>-52</sup>  | 4.5<br>× 10 <sup>-52</sup>  | 2.3<br>× 10 <sup>-21</sup> | 6.2<br>× 10 <sup>-52</sup> | 1.9<br>× 10 <sup>-21</sup> | 6.3<br>× 10 <sup>-52</sup> | 6.1<br>× 10 <sup>-52</sup> | 5.7<br>× 10 <sup>-52</sup> |                            |                            |
|                   | REF <sub>UBS</sub> | 7.6<br>× 10 <sup>-02</sup> | 2.8<br>× 10 <sup>-14</sup> | 2.5<br>× 10 <sup>-37</sup> | 1.8<br>× 10 <sup>-05</sup> | 1.1<br>× 10 <sup>-02</sup> | 6.1<br>× 10 <sup>-01</sup> | 3.4<br>× 10 <sup>-01</sup> | 1.8<br>× 10 <sup>-22</sup> | 2.8<br>× 10 <sup>-03</sup> | 5.0<br>× 10 <sup>-22</sup>  | 1.0<br>× 10 <sup>-34</sup>  | 9.4<br>× 10 <sup>-22</sup> | 3.0<br>× 10 <sup>-22</sup> | 8.4<br>× 10 <sup>-22</sup> | 6.5<br>× 10 <sup>-22</sup> | 3.3<br>× 10 <sup>-22</sup> | 3.5<br>× 10 <sup>-22</sup> | 2.0<br>× 10 <sup>-15</sup> |                            |
|                   | VSL <sub>UBS</sub> | 7.7<br>× 10 <sup>-15</sup> | 1.5<br>× 10 <sup>-03</sup> | 1.0<br>× 10 <sup>-37</sup> | 6.7<br>× 10 <sup>-16</sup> | 1.1<br>× 10 <sup>-05</sup> | 9.2<br>× 10 <sup>-44</sup> | 2.2<br>× 10 <sup>-05</sup> | 1.9<br>× 10 <sup>-31</sup> | 1.1<br>× 10 <sup>-31</sup> | 3.4<br>× 10 <sup>-05</sup>  | 5.6<br>× 10 <sup>-05</sup>  | 2.9<br>× 10 <sup>-05</sup> | 3.5<br>× 10 <sup>-05</sup> | 2.2<br>× 10 <sup>-05</sup> | 3.2<br>× 10 <sup>-05</sup> | 6.0<br>× 10 <sup>-05</sup> |                            |                            |                            |
|                   | i                  |                            |                            |                            |                            |                            |                            |                            |                            |                            |                             |                             |                            |                            |                            |                            |                            |                            |                            |                            |
| PM <sub>2.5</sub> | REF <sub>TRS</sub> | 4.2<br>× 10 <sup>-06</sup> | 3.0<br>× 10 <sup>-07</sup> | 3.5<br>× 10 <sup>-27</sup> | 2.5<br>× 10 <sup>-30</sup> | 3.1<br>× 10 <sup>-27</sup> | 8.7<br>× 10 <sup>-01</sup> | 4.9<br>× 10 <sup>-01</sup> | 1.5<br>× 10 <sup>-07</sup> | 6.9<br>× 10 <sup>-07</sup> | 3.2<br>× 10 <sup>-07</sup>  | 1.0<br>× 10 <sup>-07</sup>  | 3.9<br>× 10 <sup>-07</sup> | 2.1<br>× 10 <sup>-07</sup> | 1.1<br>× 10 <sup>-07</sup> | 1.1<br>× 10 <sup>-07</sup> | 5.0<br>× 10 <sup>-07</sup> | 1.4<br>× 10 <sup>-14</sup> | 4.0<br>× 10 <sup>-14</sup> |                            |
|                   | VSL <sub>TRS</sub> | 3.4<br>× 10 <sup>-09</sup> | 3.4<br>× 10 <sup>-01</sup> | 2.4<br>× 10 <sup>-33</sup> | 1.0<br>× 10 <sup>-03</sup> | 3.0<br>× 10 <sup>-22</sup> | 8.2<br>× 10 <sup>-11</sup> | 1.1<br>× 10 <sup>-11</sup> | 8.2<br>× 10 <sup>-18</sup> | 3.2<br>× 10 <sup>-18</sup> | 2.5<br>× 10 <sup>-02</sup>  | 1.6<br>× 10 <sup>-02</sup>  | 4.1<br>× 10 <sup>-02</sup> | 4.1<br>× 10 <sup>-02</sup> | 2.8<br>× 10 <sup>-02</sup> | 4.5<br>× 10 <sup>-02</sup> | 9.2<br>× 10 <sup>-02</sup> | 9.7<br>× 10 <sup>-12</sup> |                            |                            |
|                   | REF <sub>UBS</sub> | 6.0<br>× 10 <sup>-05</sup> | 1.4<br>× 10 <sup>-01</sup> | 2.1<br>× 10 <sup>-32</sup> | 3.5<br>× 10 <sup>-19</sup> | 3.6<br>× 10 <sup>-19</sup> | 3.2<br>× 10 <sup>-19</sup> | 4.1<br>× 10 <sup>-19</sup> | 3.0<br>× 10 <sup>-24</sup> | 3.0<br>× 10 <sup>-06</sup> | 5.0<br>× 10 <sup>-04</sup>  | 1.7<br>× 10 <sup>-04</sup>  | 1.6<br>× 10 <sup>-04</sup> | 1.5<br>× 10 <sup>-04</sup> | 1.7<br>× 10 <sup>-04</sup> | 1.9<br>× 10 <sup>-04</sup> | 9.6<br>× 10 <sup>-05</sup> | 2.6<br>× 10 <sup>-04</sup> | 2.6<br>× 10 <sup>-04</sup> |                            |
|                   | VSL <sub>UBS</sub> | 1.8<br>× 10 <sup>-10</sup> | 3.4<br>× 10 <sup>-02</sup> | 1.5<br>× 10 <sup>-37</sup> | 1.6<br>× 10 <sup>-30</sup> | 4.9<br>× 10 <sup>-06</sup> | 4.4<br>× 10 <sup>-22</sup> | 1.9<br>× 10 <sup>-22</sup> | 4.0<br>× 10 <sup>-24</sup> | 9.4<br>× 10 <sup>-06</sup> | 1.7<br>× 10 <sup>-06</sup>  | 6.7<br>× 10 <sup>-06</sup>  | 1.4<br>× 10 <sup>-06</sup> | 7.9<br>× 10 <sup>-06</sup> | 1.1<br>× 10 <sup>-06</sup> | 5.9<br>× 10 <sup>-06</sup> | 3.5<br>× 10 <sup>-06</sup> | 1.4<br>× 10 <sup>-06</sup> | 6.7<br>× 10 <sup>-13</sup> |                            |
|                   | i                  |                            |                            |                            |                            |                            |                            |                            |                            |                            |                             |                             |                            |                            |                            |                            |                            |                            |                            |                            |
| PM <sub>10</sub>  | REF <sub>TRS</sub> | 2.2<br>× 10 <sup>-05</sup> | 6.6<br>× 10 <sup>-03</sup> | 2.1<br>× 10 <sup>-35</sup> | 1.0<br>× 10 <sup>-53</sup> | 9.0<br>× 10 <sup>-03</sup> | 1.4<br>× 10 <sup>-01</sup> | 7.0<br>× 10 <sup>-04</sup> | 5.4<br>× 10 <sup>-09</sup> | 3.4<br>× 10 <sup>-06</sup> | 1.6<br>× 10 <sup>-07</sup>  | 8.0<br>× 10 <sup>-07</sup>  | 1.5<br>× 10 <sup>-07</sup> | 3.1<br>× 10 <sup>-07</sup> | 8.8<br>× 10 <sup>-07</sup> | 6.0<br>× 10 <sup>-07</sup> | 9.0<br>× 10 <sup>-16</sup> | 7.5<br>× 10 <sup>-16</sup> |                            |                            |
|                   | VSL <sub>TRS</sub> | 9.1<br>× 10 <sup>-07</sup> | 7.8<br>× 10 <sup>-01</sup> | 3.2<br>× 10 <sup>-09</sup> | 1.8<br>× 10 <sup>-03</sup> | 1.0<br>× 10 <sup>-03</sup> | 1.8<br>× 10 <sup>-03</sup> | 8.8<br>× 10 <sup>-03</sup> | 5.7<br>× 10 <sup>-24</sup> | 2.3<br>× 10 <sup>-10</sup> | 8.1<br>× 10 <sup>-12</sup>  | 7.4<br>× 10 <sup>-01</sup>  | 1.7<br>× 10 <sup>-01</sup> | 6.3<br>× 10 <sup>-01</sup> | 6.1<br>× 10 <sup>-01</sup> | 5.4<br>× 10 <sup>-01</sup> | 3.6<br>× 10 <sup>-01</sup> | 2.3<br>× 10 <sup>-01</sup> | 7.3<br>× 10 <sup>-01</sup> |                            |
|                   | REF <sub>UBS</sub> | 2.1<br>× 10 <sup>-11</sup> | 9.3<br>× 10 <sup>-03</sup> | 1.4<br>× 10 <sup>-35</sup> | 1.0<br>× 10 <sup>-53</sup> | 1.9<br>× 10 <sup>-03</sup> | 2.1<br>× 10 <sup>-03</sup> | 2.5<br>× 10 <sup>-03</sup> | 1.9<br>× 10 <sup>-03</sup> | 6.3<br>× 10 <sup>-03</sup> | 1.5<br>× 10 <sup>-03</sup>  | 3.3<br>× 10 <sup>-03</sup>  | 1.4<br>× 10 <sup>-03</sup> | 5.6<br>× 10 <sup>-03</sup> | 1.1<br>× 10 <sup>-03</sup> | 6.2<br>× 10 <sup>-03</sup> | 5.0<br>× 10 <sup>-03</sup> | 7.1<br>× 10 <sup>-03</sup> | 2.8<br>× 10 <sup>-02</sup> |                            |
|                   | VSL <sub>UBS</sub> | 2.2<br>× 10 <sup>-13</sup> | 2.5<br>× 10 <sup>-03</sup> | 4.5<br>× 10 <sup>-33</sup> | 1.2<br>× 10 <sup>-12</sup> | 1.6<br>× 10 <sup>-27</sup> | 8.9<br>× 10 <sup>-19</sup> | 6.5<br>× 10 <sup>-19</sup> | 4.0<br>× 10 <sup>-52</sup> | 2.0<br>× 10 <sup>-08</sup> | 10.9<br>× 10 <sup>-08</sup> | 10.9<br>× 10 <sup>-08</sup> | 2.2<br>× 10 <sup>-08</sup> | 8.1<br>× 10 <sup>-08</sup> | 2.5<br>× 10 <sup>-08</sup> | 4.8<br>× 10 <sup>-08</sup> | 1.8<br>× 10 <sup>-08</sup> | 1.5<br>× 10 <sup>-08</sup> |                            |                            |
|                   | i                  |                            |                            |                            |                            |                            |                            |                            |                            |                            |                             |                             |                            |                            |                            |                            |                            |                            |                            |                            |

**Commented [RP3]:** Updated to be a table (not a figure)

### 3.4 Comparison of the performance of NO<sub>2</sub> and O<sub>3</sub> sensors before and after firmware update

As discussed in section 2.2, the firmware for the NO<sub>2</sub> and O<sub>3</sub> sensors changed during the course of the measurements (on 25/08/2022 for the NO<sub>2</sub> and on 26/01/2023 for the O<sub>3</sub> sensors), providing an opportunity to test how this can affect the measurements.

The target diagrams provided in Fig. 6 show the performance of these sensors (Fig. 6a for NO<sub>2</sub> and Fig. 6b for

O<sub>3</sub>) against measurements by the respective reference instruments before and after the firmware updates.

The distance of each point from the centre of the circle corresponds to the normalized, by the standard deviation of the reference measurements, RMSE (nRMSE) described in section 2.3. As shown in the target diagrams, the performance of the Vaisala AQT530 NO<sub>2</sub> and O<sub>3</sub> sensors at both stations was improved after the firmware updates, as indicated by the lower nRMSE values. More specifically, the magnitude of the nRMSE vector decreased by 63.5% (VSL<sub>TRS</sub>) and 73.4% (VSL<sub>UBS</sub>) for the NO<sub>2</sub> sensors, and by 57.9% (VSL<sub>TRS</sub>) and 27.5% (VSL<sub>UBS</sub>) for the O<sub>3</sub> sensors following their firmware updates. Regarding the MBE, there was an improvement of 68.0% (VSL<sub>TRS</sub>) and 70.2% (VSL<sub>UBS</sub>) for the NO<sub>2</sub> sensors, and of 58.6% (VSL<sub>TRS</sub>) and 356.3% (VSL<sub>UBS</sub>) for the O<sub>3</sub> sensors. The CRMSE also improved by 57.0% (VSL<sub>TRS</sub>) and 60.6% (VSL<sub>UBS</sub>) for the NO<sub>2</sub> sensors and 61.19% (VSL<sub>TRS</sub>) and 71.6% (VSL<sub>UBS</sub>) for the O<sub>3</sub> sensors. Determining whether this overall performance improvement

enables the sensors to capture the spatial and temporal differences discussed above, would require a more extended study in  
which the new firmware is used for at least 12 months with a new set of sensors, considering their limited lifespan.

Figure 6: Target diagrams showing the bias and variance of the NO<sub>2</sub> (a) and the O<sub>3</sub> (b) LCSs employed in the AQT530 monitors against reference measurements before (open symbols) and after (solid symbols) their firmware updates (on 25/08/2022 for the NO<sub>2</sub> and on 26/01/2023 for the O<sub>3</sub> sensor). If the variance of the residuals (VSL-Ref) is smaller than the variance of the reference measurements, the data points should fall within the blue circle.

#### 4 Conclusions

We have carried out air quality monitoring measurements using two Vaisala AQT530 monitors and reference instruments at a traffic and an urban background station in Nicosia, Cyprus, and investigated if the LCSs employed in the former can capture the spatial and temporal variabilities similarly to the latter. Initial measurements where both Vaisala AQT530 monitors were  
collocated with reference instruments at the traffic station showed that only the CO and PM<sub>10</sub> measurements exhibit a high enough correlation and agreement with the reference measurements, and a good sensor-to-sensor reproducibility.

Analysis of the reference measurements shows that the mean concentrations of the pollutants at the two stations, over the entire study period and for each month separately, were statistically significantly different at a 95% confidence interval. On an hourly basis, the reference measurements also showed statistically significant differences for certain hours of the day at the two  
stations. The respective Vaisala AQT530 low-cost measurements for CO and PM<sub>10</sub> were able to capture the significance of the spatial gradients between the two stations for the entire study period and on a monthly basis, with the exceptions of a few months depending on the sensor. On daily (workdays or weekends) or hourly basis, the ability of the Vaisala AQT530 LCS CO and PM<sub>10</sub> measurements to capture the spatial differences was higher in cases where the difference in the concentration of the pollutants at the two stations was higher (i.e., > 80 ppb for CO and > 10 µg/m<sup>3</sup> for PM<sub>10</sub>).

Regarding the temporal (i.e., monthly) variations, the CO and PM Vaisala AQT530 LCSs captured the month-to-month reference trend over the entire period, while the NO<sub>2</sub>, NO and O<sub>3</sub> LCSs did not, mainly due to their sensitivity on the environmental conditions. PM and marginally the NO and NO<sub>2</sub> LCSs capture the reference month-to-month differences in more than 70% of the cases. However, considering that the latter fail to reproduce the overall trends, reporting values that are significantly different compared to the reference instruments, they can be excluded, leaving only the PM sensor as the most  
reliable indicator of temporal variabilities at the same station.

Overall, among all Vaisala AQT530 LCSs, the ones measuring PM appear to capture better than the others both the temporal and spatial resolutions (which is not a surprise given the more robust operating principle compared to the gas sensors) despite their relatively high cut-off diameter, while the CO sensors can be used to capture effectively mainly spatial differences. The CO LCSs managed to capture the monthly and diurnal trends/variabilities regardless of the environmental conditions, while  
the rest of the low-cost gas sensors appear to report measurements that are comparable to those from the reference instruments only at lower temperatures (< 20 °C) and higher RH (> 55%).

#### **Data Availability**

Papaconstantinou, R. (2024). Assessing Spatial and Temporal Urban Air Quality Variabilities with the Vaisala AQT530 Monitor [Data set]. Zenodo. <https://doi.org/10.5281/zenodo.13985490>

**Authorship contribution**

R. Papaconstantinou: Writing – review & editing, Writing – original draft, Visualization, Validation, Software, Methodology, Investigation, Formal analysis, Data curation. S. Bezantakos: Writing – review & editing, Supervision, Conceptualization. M. Pikridas: Writing – review & editing, Resources. M. Parolin: Writing – review & editing, Resources. M. Stylianou: Resources. C. Savvides: Resources. J. Sciare: Writing – review & editing, Funding acquisition. G. Biskos: Writing – review & editing,  
Supervision, Conceptualization.

#### **Financial support**

This project has received funding from the European Union’s Horizon 2020 Research and Innovation Programme under Grant Agreement No. 856612 (EMME-CARE Project) and the Cyprus Government.

#### **Competing interests**

The authors declare that they have no conflict of interest.

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
