# Peer review of "Comparing Spatial and Temporal Variabilities between the Vaisala AQT530 Monitor and Reference Measurements"

_EGUsphere, 2025_

## Referee Comment (RC1)

"Assessing Spatial and Temporal Urban Air Quality Variabilities with the Vaisala AQT530 Monitor"

**General Comments**

The manuscript presents the assessment of the Vaisala AQT530 Monitor in Cyprus. It is a long-term study (19 months) of two units deployed in an urban background and traffic location which has reference grade instrumentations for the co-monitored pollutants (CO, NO, $NO_2$, $O_3$, $PM_{2.5}$ and $PM_{10}$). The authors showed that only the CO and to lesser extent PM readings had reasonable agreement with the reference monitoring data in terms of temporal (diurnal, seasonal) and spatial (background vs traffic) variability. They attributed the poor performance of the Vaisala AQT530 for the other species to the impact of environmental factors like RH and temperature. The strength of this work is the detailed statistical analysis carried out by the authors during the assessment of the AQT530.

**Specific comments**

The reviewer if of the opinion that this work shows the performance limitation of the commercial LCS device that the authors used rather than a general conclusion of the limitations of LCS application in ambient AQ monitoring because there are large body of work that shows this, the authors need to capture this in some form in their conclusions.

**Technical corrections**

The abbreviation 'cf.' is incorrectly being used throughout the manuscript. It means "to compare to" and I think the authors are using it to mean "see". Authors need to correct this.

Authors sometimes also make statements that are not really justified in the first instance but are subsequently justified later on in the manuscript. For instance, in the final paragraph on page 6, the authors introduced periods classified as dust and non-dust period without justification or any citation to back this up, but did subsequently add a citation on page 8 second sentence when discussing the dust event. I would suggest the authors look into instances of this and make corrections.

On page 8, the reviewer also struggle to agree with authors the general conclusion that systematic bias in the LCS PM reading between the background site (UBS) and the traffic site (TRS) is mainly due to the former having less particles with diameter < 600 nm compared to the traffic site and the inability of the Vaisala unit to detect below this cutoff. This conclusion cannot explain why the LCS UBS is still significantly biased high compared to the TRS during dust episodes (Fig. 4e) when the PM is expected to be dominated by large diameter sized particles

Page 8 first paragraph " …. PM10 concentrations, however, are generally higher at the TRS compared to the UBS, as indicated by the reference instruments (Fig. 3g), which is in contrast to what the LCS measurements indicate (Fig. 3h)" Figures 3g and 3h are scatter plots for CO and $NO_2$ respectively that have no relation to $PM_{10}$. Authors need to correct this.

Page 8 second paragraph " …. while those measured by the reference instruments were below 6–10 µg/m3" this statement is wrong. The difference for the reference instruments is never negative between the TRS and USB site as shown in Fig. 4f (solid line).

Authors need to add legends to Fig 4, Fig 5, Fig S14 and Fig S15 to help the readers.

Suggest the authors annotate Fig S2 to show when the firmware was changed for the $NO_2$ and $O_3$ sensors.

Table 4 appears to be an "image" and the font size is too small making it difficult to read. I suggest presenting this as an actual table and to increase the font size.

---

## Author Comment (AC1)

**Specific comments**

**RC1**: The reviewer is of the opinion that this work shows the performance limitation of the commercial LCS device that the authors used rather than a general conclusion of the limitations of LCS application in ambient AQ monitoring because there are large body of work that shows this, the authors need to capture this in some form in their conclusions.

**AC:** We agree that our findings primarily reflect the performance limitations of the specific commercial LCS device used in our study, rather than representing a general limitation of all LCS applications in ambient air quality monitoring. In response, we revised the conclusion section to clearly state that our observations are device-specific.

**Technical corrections**

**RC1**: The abbreviation 'cf.' is incorrectly being used throughout the manuscript. It means "to compare to" and I think the authors are using it to mean "see". Authors need to correct this.

**AC:** We acknowledge the incorrect use of the abbreviation "cf." in the manuscript. We revised the text to replace it with the appropriate term (e.g., "see") where necessary.

**RC1**: Authors sometimes also make statements that are not really justified in the first instance but are subsequently justified later on in the manuscript. For instance, in the final paragraph on page 6, the authors introduced periods classified as dust and non-dust period without justification or any citation to back this up, but did subsequently add a citation on page 8 second sentence when discussing the dust event. I would suggest the authors look into instances of this and make corrections.

**AC:** We thank Referee 1 for pointing this out. In the updated version of the manuscript, we have changed the final paragraph on page 6 to:

"We should highlight here that the agreement between PM LCSs and reference instruments improved significantly during periods with dust events, a phenomenon that occurs with higher frequency and intensity during spring and autumn in the region (Yukhymchuk et al. 2022, Kezoudi et al. 2021)."

and page 8 second sentence to:

"The striking peak observed in April 2022 is due to a strong dust event as discussed above in section 3.1".

We have also gone through the manuscript and identified a few more points having the same issues and did the necessary corrections.

**RC1**: On page 8, the reviewer also struggle to agree with authors the general conclusion that systematic bias in the LCS PM reading between the background site (UBS) and the traffic site (TRS) is mainly due to the former having less particles with diameter < 600 nm compared to the traffic site and the inability of the Vaisala unit to detect below this cutoff. This conclusion cannot explain why the LCS UBS is still significantly biased high compared to the TRS during dust episodes (Fig. 4e) when the PM is expected to be dominated by large diameter sized particles

**AC:** We agree that a difference remains between the UBS and TRS sensors during dust events (Fig. 4e). However, this does not contradict the claim that the presence of smaller particles is higher at the TRS. To clarify that, we added a supplementary figure (Figure S6 in the updated supplement) showing that, even during the dust event, $PM_{2.5}$ concentrations are consistently higher at the TRS compared to UBS. This figure highlights that the TRS station systematically records higher concentrations of smaller particles ($PM_{2.5}$). That said, it does not exclude the fact that TRS also experiences more coarse particles than UBS during dust events (see subplot c in the figure below).

[Figure]

*Figure 1: Time series of 1-hour averaged $PM_{2.5}$ and PM coarse concentrations measured by the reference instruments at TRS and UBS. The differences in $PM_{2.5}$ and PM coarse between the two reference measurements (i.e., $REF_{TRS} - REF_{UBS}$) are also plotted. The solid black line represents zero. Data between the red dashed lines indicate a major dust event in April 2022.*

**RC1**: *Page 8 first paragraph " …. PM10 concentrations, however, are generally higher at the TRS compared to the UBS, as indicated by the reference instruments (Fig. 3g), which is in contrast to what the LCS measurements indicate (Fig. 3h)" Figures 3g and 3h are scatter plots for CO and NO2 respectively that have no relation to PM10. Authors need to correct this.*

**AC:** We thank the referee for seeing this error. This was indeed a typo. We will correct the text to refer to the appropriate figures, changing "Fig. 3g" to "Fig. 4d" and "Fig. 3h" to "Fig. 4e," which correctly correspond to the $PM_{10}$ data being discussed.

**RC1**: *Page 8 second paragraph " …. while those measured by the reference instruments were below 6–10 µg/m3" this statement is wrong. The difference for the reference instruments is never negative between the TRS and USB site as shown in Fig. 4f (solid line).*

**AC:** We acknowledge that the word 'below' may have misled the meaning of this sentence. We revised the sentence to:

"The PM concentration differences between the two stations in these cases, as measured by the LCSs, were less than 4 µg/m$^3$ while those measured by the reference instruments were less than 10 µg/m$^3$."

**RC1**: *Authors need to add legends to Fig 4, Fig 5, Fig S14 and Fig S15 to help the readers.*

**AC:** We agree and added appropriate legends to all these figures.

**RC1**: Suggest the authors annotate Fig S2 to show when the firmware was changed for the NO2 and O3 sensors.

**AC:** The point is well taken. We have updated Fig. S2 to indicate the times when the firmware was changed for the NO$_2$ and O$_3$ sensors.

**RC1**: Table 4 appears to be an "image" and the font size is too small making it difficult to read. I suggest presenting this as an actual table and to increase the font size.

**AC:** We updated the table accordingly.

---

## Author Comment (AC2)

**Major comments**

**RC2**: The authors compare the performance of two LCS and a reference monitor- specifically they evaluate temporal and spatial trends and find somewhat of a correlation. Why was no calibration model applied, despite the Introduction stating that such models are necessary?

**AC:** We thank the referee for this point. The Vaisala AQT530 monitors include proprietary calibration algorithms embedded in their firmware. Those are designed to adjust raw sensor responses in real time, compensating for the impact of ambient conditions and aging of the sensor elements. Our goal was to evaluate the performance of these sensors as provided by the manufacturer, and to determine whether they can reliably capture spatial and temporal variability in pollutant concentrations without further post-corrections. By analysing the sensors, which come calibrated by the manufacturer, we aim to determine their real-world performance. To make this point clear we have updated the manuscript in several places. For example lines 88 – 90 in the updated version of the manuscript now read: "The Vaisala Air Quality Transmitter (AQT530) is one of the commercially available cost-effective air quality monitors that incorporates proprietary algorithms designed to compensate effects related to variable environmental conditions and sensor aging (AQT530; 2023).".

**RC2**: Why were the sensors not co-located with the reference monitor for a longer time to compare the two? Doing so would provide much more definitive results than the authors current approach.

**AC:** We understand the point Referee 2 makes here. The scope of this study is to assess the capability of the Vaisala AQT sensors in capturing spatial and/or temporal variabilities of air pollutants within the urban environment. Doing so in a reliable way, however, requires to first verifying that the two AQT units provided consistent measurements when exposed to identical environmental and pollution conditions. For this reason, both AQTs were initially co-located with reference instrumentation at a single site for a period of one week. Despite that one week can be considered a short period of time for such an inter-comparison, the conditions encountered covered a wide range of meteorological conditions and pollutant concentrations. To be more specific, during the period we co-located the sensors, we collected 1897 data points, capturing temperature variations between 10 and 33 °C and relative humidity levels from 11 to 95% (as illustrated in figure below). Additionally, the corresponding ranges of pollutant concentrations encountered were very similar to the ones measured at the two different locations throughout the 19-month campaign (see Figure 3 of the main manuscript and Figure S1 of the Supplement). During the inter-comparison period the measurements of the two AQT monitors were identical for CO and $PM_{10}$, exhibiting a slope and an $R^2$ of ca. 1.0, upon comparison of the two monitors (see Figure 2 in the manuscript). The measurements of NO, $NO_2$, $O_3$ and $PM_{2.5}$ reported by the two AQT monitors showed differences (see Figure 2), the significance of which was further evaluated by WRS tests. These tests indicated that the concentrations of CO, $PM_{10}$ and $NO_2$ reported by the two AQT monitors did not exhibit significant statistical differences (within 5%) for each other (see lines 220 – 226 in the manuscript). Based on the results of the inter-comparison tests we used for our study the CO and $PM_{10}$ measurements of the AQT monitors.

We would also like to clarify that although the two AQTs were deployed at separate urban monitoring locations after the initial co-location, each remained co-located with a reference-grade instrument at its respective site for the remainder of the monitoring campaign. This approach ensured reliability and robustness of the comparative analyses presented in the study.

All of the above are now more clearly written in section 3.1 of the manuscript.

[Figure]

*Figure 1: Time series of the temperature and RH measurements recorded during the co-location week.*

**RC2**: The authors results are not novel. I would advise the authors to use the sensors to investigate a specific air quality issue, instead of merely providing correlations between these sensors

**AC:** We believe that the results of this work are very important. Before using those sensors to investigate any air quality issues, we need to assess their performance, and doing so, we need to check their ability to capture temporal and spatial variabilities. Specifically, our work assesses whether the variability in the concentration of different air pollutants shown by the Vaisala AQT530 measurements can be attributed to actual spatial and/or temporal differences within the complex urban environment or is an artefact related to the limitations of the current technology used in low/medium cost and portable sensors. To the best of our knowledge, and despite of its importance, this kind of assessment is missing in the literature. Understanding how well (or not) these sensors reflect intra-urban differences; despite their limitations, contributes meaningful insights for all specific AQ applications, like practical air quality management, hot-spot recognition, emission sources activity time and location profile, exposure etc. We believe this perspective adds relevance beyond correlations, offering guidance on real-world use of the AQT530 in city-scale environmental monitoring. The importance of our study is now better emphasized at the end of the introduction.

**RC2**: In addition, a lot of key details are in the SI such as details of the WRS test.

**AC:** We chose to include supporting information and details in the supplement, such as the detailed description and equation of the WRS test, which is a well-known statistical test. The same is true for the p-values resulting from the test (also included in the supplement) which only show (and support) if the null hypothesis of the tests were accepted or rejected at the significance level used in our study (i.e., 5%). By including such results in the manuscript would have made the presentation less streamlined and potentially harder to follow.

**Minor comments**

**RC2**: In section 2.2 was there any reason the Vaisala monitors were selected? Have these sensors been evaluated by the EU JRC or CARB or US EPA? Can the authors report what these organizations have found during their lab calibrations/field tests?

**AC:** The Vaisala AQT530 is a commercially available monitor but has only been used in a limited number of studies to date. Therefore, evaluating its performance is both timely and necessary. To the best of our knowledge, there is no publicly available information on whether the Vaisala AQT530 has been evaluated by the European Commission's Joint Research Centre (JRC), the California Air Resources Board (CARB), or the U.S. Environmental Protection Agency (EPA). However, it has undergone evaluation in 2022 by the South Coast Air Quality Management District (AQMD) in California, and specifically at the Air Quality Sensor Performance Evaluation Center (AQ-SPEC). Key results from the tests conducted by AQ-SPEC are now presented in the revised manuscript (see lines 90-100).

Key performance indicators of the field performance evaluation from AQ-SPEC were also briefly compared with our results in section 3.1 of the revised manuscript (see lines 264 – 270).

**RC2**: In section 2.3 can the authors provide information on the manufacturer specifications

**AC:** To address that, we have included the key specifications of all the sensors used in the monitor in section 2.2 (see Table 2) of the revised manuscript version.

---

## Referee Report (RR1)

The authors have addressed all my concerns